# A Wind-Turbine-Tower-Climbing Robot Prototype Operating at Various Speeds and Payload Capacity: Development and Validation

**Kathleen Ebora Padrigalan** and **Jui-Hung Liu** *

Department of Mechanical Engineering, Southern Taiwan University of Science and Technology, No. 1, Nantai Street, Yongkang District, Tainan City 710301, Taiwan
* Correspondence: dofliu@stust.edu.tw

**Abstract:** The development of control technology on wind turbine application robots has played an integral role in facilitating the digitization of inspection and maintenance in the wind energy industry. This paper presents a wind-turbine-climbing robot that determines the service lifespan of the wind turbine components subject to its payload capacity. The model has four rubber wheels, as the driving mechanism for its locomotion is being supported by a Bowden cable as a winding mechanism for its adhesion. The design further incorporates an Arduino microcontroller, distance sensors, motors, and a step motor to form its electromechanical structure. The overall capability of the robot has been analyzed through its kinematics and dynamics. Practical indoor experiments using a wind turbine tower mockup have been conducted for the validation of the various speeds and payload capacity of the prototype. The results indicate the effectiveness of its driving and winding mechanism to climb at the various speeds and with or without a payload. The advantage of the operations of its mechanism conformed with the wind turbine application robots.

**Keywords:** wind-turbine-tower-climbing robots; locomotion; adhesion; winding mechanism; driving mechanism





## 1. Introduction

The advancement of control technologies and the intelligent research on climbing robots [1] receive extensive interest in the academe–industry community to promote scientific discipline, research innovation and technological developments in the field of wind turbine application robots [2–4]. Wind-turbine-climbing robots facilitate the digitization of inspection and maintenance operation to prolong the service lifespan and to avoid unexpected external failures of the wind turbine parts [5]. This includes corrosion, cracks, paint peel-off, material degradation, lightning strike damage and other physical defects [3,6], as listed in Table 1. These conditions seriously damage the physical components of the wind turbine, leading to possible major issues, or even the worst-case scenarios. The corrosion of wind turbines is a vexed issue in its steel structure, which may cause dependency on different characteristics attacked by high or unwanted levels of humidity, on the inside and outside of the turbine system [7]. Wind turbine tower cracks could result in catastrophic failures due to extreme loads on the tower, improper manufacturing, poor material and structural design and excessive environmental factors [8,9]. Such cracks were detected in its circumferential welded joints between the lower rings and flanges connecting the tower and its base foundations [10]. The peeling of paints, caused by wind, raindrops, airborne sand, temperature changes, and acidity from bug carcasses [11], could damage the adhesive layer and the bond between the skin layer [12] and the surface of the wind turbine parts. The most prevalent material degradation is fatigue, due in part to the very large number of cycles and types of internal interfaces in the composite material components that are potential weak links. Offshore turbines, with their greater exposure to higher

levels of moisture and salt, can suffer from material degradation [13]. Lightning strike is a major challenge due to wind turbines being tall structures, especially when placed in flat planes [14]. Two types of lightning may occur: (1) a downward initiated lightning, which starts in a thunderstorm and propagates downward, attaching to the turbine tip, and (2) an upward lightning, which happens when the turbine is very tall and itself starts to generate lightning. Other surface defects are caused by erosion (rain erosion, sand, and hail) and small object impacts [12]. Overall, these external defects severely shorten the wind turbines' service life and restrict the cost per MW. Thus, this study aims to resolve the major issues related to the short lifespan of the wind turbines components and to improve the overall efficiency of wind turbine components, optimize manpower, identify issues early on and schedule maintenance to prevent unnecessary costs and catastrophic external wind turbine failures in the future through the benefits of a climbing robot. This suggests that the study of the wind-turbine-climbing robots to carry out surface maintenance in the different wind turbine components is extremely necessary and is deemed to become a common activity in the succeeding years, as the number of installed wind turbines is rising globally, necessitating further research for this topic.

**Table 1.** Types of physical defects occurring in the structure of wind turbine.

| WT Physical Defects | Description | References |
|---|---|---|
| Corrosion | A different characteristics attack by a high or unwanted levels of humidity inside and outside the wind turbine system. | [7] |
| Cracks | It is due to extreme loads on the tower, improper manufacturing, poor material and structural design and excessive environmental factors. | [8,9] |
| Paint peel-off | The paint peel-off caused by the elements of wind, raindrops, airborne sand, temperature changes, and acidy from bug carcasses. | [11] |
| Material Degradation | It is due in part to the very large number of cycles and types of internal interfaces in the composite material components that are potential weak links; with their greater exposure to higher levels of moisture and salt. | [13] |
| Lightning Strike | It is due to tall structures that are attractive to lightning, especially when located in flat planes. | [14] |
| Other surface defects | This is caused by erosion (rain erosion, sand, and hail), or small object impacts. | [12] |

Studies on wind-turbine-climbing robots mostly focused on two important mechanisms: the adhesion and locomotion [15–17]. Adhesion is the mechanism that makes firm contact to the surface/object without slipping. Various adhesion mechanisms have been identified in the application of wind turbines with its prototype models. Among these adhesion mechanisms are magnetic adhesion [2,17,18], vacuum/air suction adhesion [19,20], ropes/rubber bands traction [21], and mechanical-spring traction [22,23]. Locomotion is the mechanism that enables the robot to move unbounded throughout its environments. Several types in terms of locomotion mechanism are being classified on wind-turbine-climbing robots, such as wheeled [14,23], tracked [2,15], and legged [19]. Depending on the features of structural components, robots have a variety of climbing mechanisms. Some of the recent studies on wind-turbine-climbing robots' application [2,18–21,23–25] use experimental prototypes. One of the applications of a magnetic climbing robot is for the inspection and maintenance of a wind tower, using magnets to keep it in contact with the tower [2]. A two-footed climbing robot that uses vacuum suckers was used to adhere to the rotor blade or to the tower [20], the Lego-based robot for the inspection of wind turbine parts with diagonal ropes and rubber bands was employed for the connection to the tower [21], and the climbing ring robot for offshore wind turbines can climb around cylindrical towers [23]. After reviewing the wind-turbine-climbing robot with its two important mechanisms, the functionality of its locomotion and adhesion, this study proposed four rubber wheels that perform locomotion and a winding mechanism that provides enough adhesion to the wind turbine tower.

The paper aims to develop a wind-turbine-tower-climbing robot with a high payload capacity to install additional equipment and to withstand the impact of wind disturbances. The locomotion mechanism is a wheel-based method for a quicker and more robust climb. The four rubber wheels create enough friction as it rolls over the tower's surface, preventing it from slipping. By combining rubber wheels as its locomotion mechanism and a winding mechanism through a Bowden cable force, one can maintain its position on the tower surface without it slipping to the tower base and increase the contact on the tower surface to achieve a high payload capacity. Concerning the robot carrying load capability, an additional component in the body frame through the aid of the caster wheel was added to optimize its stability for carrying ancillary equipment to perform specific application tasks of wind turbine parts and wind disturbances. With these goals, the carrying load was roughly more than 50% of the robot's weight that was protected from breaking or slipping. This climbing robot can perform a straight up–down movement on the tower surface. The other unique characteristic of the proposed design was that it can achieve wind gust disturbance's ability and innovative structure to estimate the robot position in the manner to maximize its capacity for climbing, which were the main drawbacks of the existing climbing robots.

This paper was organized as follows: Section 2 presents a description of the wind-turbine-tower-climbing robot, kinematic and dynamic equation of motion, control system and experimental set-up. Section 3 was devoted to the experimental results performed of the various speeds and subject to payload capacity and the indoor testing conditions, and Section 4 provided the conclusions and future works.

## 2. Materials and Methods

The Materials and Methods section highlights the wind-turbine-tower-climbing robot, with a specific description of parts, as well as a kinematic and dynamic equation of motion.

### 2.1. Climbing Robot Description

The design of the wind-turbine-tower-climbing robot is shown in Figure 1 with the following specific parts: the Power Supply, Twin Pulley, Step Motor—42HS02, Body Frame, Bowden Cable, Arduino Microcontroller, Electronic Module—Relay, Distance Sensor, Motors—CHP-36GP-Bl3650, Rubber Wheels and Caster Wheel. The current structure of wind-turbine-tower-climbing robot consists of three main parts: the driving mechanism, winding mechanism, and body frame. The driving mechanism of the robot was composed of four motorized rubber wheels, which combined the advantages of the wheel structure aligned to the robot's body frame. The robot's mobility was being driven by the high torque and high power of the DC motors with 12 V supply. The winding mechanism used a step motor with fabricated twin pulley and Bowden cable to provide the tension force that held the climbing robot to the tower surface. This mechanism adapted to the different diameters of the tower mock-up and holds slow speed adjustments of cable force onto the tower surface without falling to the ground. Through the work of the ultrasonic distance sensor attached to the robot's body frame, the cable tension force acted as an adhesion mechanism dependent on the feedback signal given by sensor to maintain a precise distance of cable to wind or unwind. The Arduino MEGA 2560, an embedded microprocessor module, integrated all the controls for driving mechanism, winding mechanism, and other feedback signals to facilitate movement in a straight up/down motion into the wind turbine tower mock-up. With this new transformative design, the robot had the potential to adapt to the wind turbine tower diameter structures and provide greater mobility to examine physical defects. Once the climbing robot was placed on the tower mock-up, the body frame was locked by the cable tension force as an adhesion mechanism of the robot. The body frame consisted of two parts, both having light structure and being easy to install and transport. The climbing robot had a caster wheel to increase the stability of the body frame position, causing the four rubber wheels to come in contact with the tower upon climbing.

It also helped the robot to perform effectivity both in driving and winding mechanism. All electronic accessories attached to its body frame are excluded from its payload capacity.

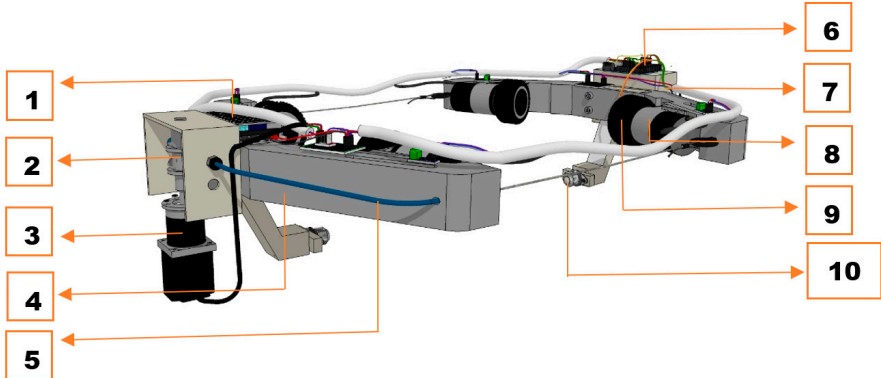

**Figure 1.** Robot Design: (**1**) Power Supply; (**2**) Twin Pulley; (**3**) Step Motor—42HS02; (**4**) Body Frame; (**5**) Bowden Cable; (**6**) Arduino Microcontroller; (**7**) Electronic Module—Relay, Distance Sensor; (**8**) Motors—CHP-36GP-Bl3650; (**9**) Rubber Wheels; and (**10**) Caster Wheel.

### 2.2. Kinematic and Dynamic Equation of Motion

In this context, the details of the wind-turbine-tower-climbing robot are depicted in a top-view position in Figure 2 for easy reference and detailed identification of robot tagging, where $W_1$ is wheel 1, $W_2$ is wheel 2, $W_3$ is wheel 3, $W_4$ is wheel 4, $CW_1$ is caster wheel 1, $CW_2$ is caster wheel 2, $BF_1$ is body frame 1 with winding mechanism and $BF_2$ is body frame 2 without winding mechanism. The robot's model is simplified in a 2D model in the $Z$-$X$ axis, as depicted in Figure 3—the model of the wind-turbine-tower-climbing robot's wheels—in which the $Z$ axis refers to the straight up–down motion of the robot; $X$ axis is the tension direction of the cable; $\varnothing_1$ and $\varnothing_3$ are an angle of line from center point of the wheel to the top wheel contact point to the tower surface; $\varnothing_2$ and $\varnothing_4$ are an angle of line from the center of the caster wheel to the center of the wheel with respect to $Z$ axis; $\varnothing_5$ is an angle from the origin center of mass with respect to the $X$ axis; $G$ is the center of the robot's body frame; $G'$ is the robot's center of mass; $A$, $B$, $C$ and $D$ are the center point of the each wheel; $E$ and $F$ are the center point of the caster wheel; $O_1$, $O_2$, $O_3$ and $O_4$ are the topmost contact point of the wheel. Other kinematic analysis could be calculated from these parameters, provided that the four wheels were in uniform rotation during the wind turbine tower climbing. Thus, the position, speed, and acceleration to the center of the tower diameter of the robot's moving wheel, the robot's center of mass and center of mass of the driving wheel and caster wheel can be determined.

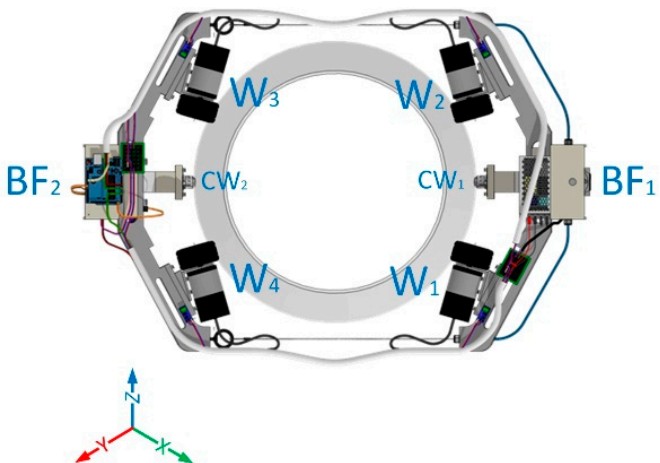

**Figure 2.** Robot's top view with tagging.

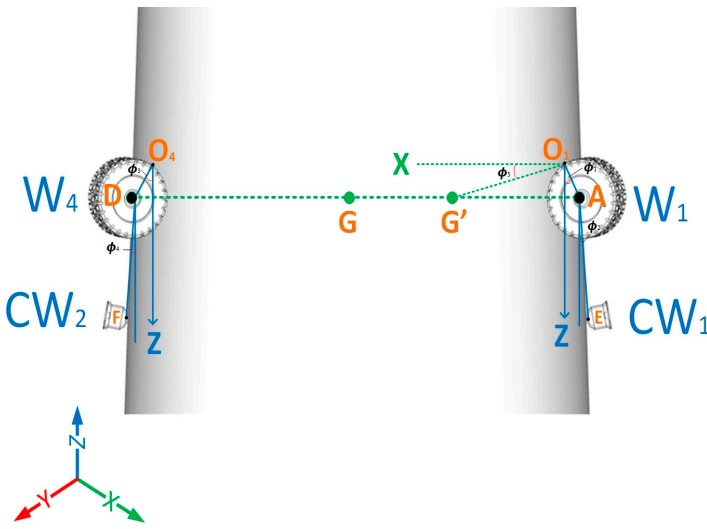

**Figure 3.** Model of the tower-climbing robot's wheel.

2.2.1. Kinematic Analysis for Wheel 1 and Caster Wheel 1

Position of $W_1$ and $CW_1$

The positions of wheel 1 and caster wheel 1 are centroid coordinates of the driving wheel that can be achieved by Equations (1)–(4):

$$Z_{W1} = L_{O_1A} \cos \varnothing_1 \tag{1}$$

$$X_{W1} = L_{O_1A} \sin \varnothing_1 \tag{2}$$

$$Z_{CW1} = Z_{W1} + L_{AE} \cos \varnothing_2 = L_{O_1A} \cos \varnothing_1 + L_{AE} \cos \varnothing_2 \tag{3}$$

$$X_{CW1} = X_{W1} + L_{AE} \sin \varnothing_2 = L_{O_1A} \sin \varnothing_1 + L_{AE} \sin \varnothing_2 \tag{4}$$

Speed of $W_1$ and $CW_1$

By calculating the first-order derivative of Equations (1)–(4), the centroid speed of wheel 1 and caster wheel 1 can be expressed as follows:

$$\dot{Z}_{W1} = -\dot{\varnothing}_1 L_{O_1A} \sin \varnothing_1 \tag{5}$$

$$\dot{X}_{W1} = \dot{\varnothing}_1 L_{O_1A} \cos \varnothing_1 \tag{6}$$

$$\dot{Z}_{CW1} = \dot{Z}_{W1} - \dot{\varnothing}_2 L_{AE} \sin \varnothing_2 = -\dot{\varnothing}_1 L_{O_1A} \sin \varnothing_1 - \dot{\varnothing}_2 L_{AE} \sin \varnothing_2 \tag{7}$$

$$\dot{X}_{CW1} = \dot{X}_{W1} + \dot{\varnothing}_2 L_{AE} \cos \varnothing_2 = \dot{\varnothing}_1 L_{O_1A} \cos \varnothing_1 + \dot{\varnothing}_2 L_{AE} \cos \varnothing_2 \tag{8}$$

Acceleration of $W_1$ and $CW_1$

By calculating the second-order derivative of Equations (5)–(8), the centroid acceleration of wheel 1 and caster wheel 1 can be expressed as follows:

$$\ddot{Z}_{W1} = -\dot{\varnothing}_1{}^2 L_{O_1A} \cos \varnothing_1 \tag{9}$$

$$\ddot{X}_{W1} = -\dot{\varnothing}_1{}^2 L_{O_1A} \sin \varnothing_1 \tag{10}$$

$$\ddot{Z}_{CW1} = \ddot{Z}_{W1} - L_{AE}\left( \dot{\varnothing}_2{}^2 \cos \varnothing_2 \right) = -\dot{\varnothing}_1{}^2 L_{O_1A} \cos \varnothing_1 - L_{AE}\left( \dot{\varnothing}_2{}^2 \cos \varnothing_2 \right) \tag{11}$$

$$\ddot{X}_{CW1} = \ddot{X}_{W1} - L_{AE}\left( \dot{\varnothing}_2{}^2 \sin \varnothing_2 \right) = -\dot{\varnothing}_1{}^2 L_{O_1A} \sin \varnothing_1 - L_{AE}\left( \dot{\varnothing}_2{}^2 \sin \varnothing_2 \right) \tag{12}$$

Equations (9)–(12) constitute the kinematic equation for the robot driving wheel and caster wheel in climbing the wind turbine tower capability.

### 2.2.2. Kinematic Analysis for Wheel 4 and Caster Wheel 2

Position of $W_4$ and $CW_2$

The position of wheel 4 and caster wheel 2 are centroid coordinate of the driving wheel that can be achieved by Equations (13)–(16):

$$Z_{W4} = L_{O_4D} \cos \varnothing_3 \tag{13}$$

$$X_{W4} = L_{O_4D} \sin \varnothing_3 \tag{14}$$

$$Z_{CW2} = Z_{W4} + L_{DF} \cos \varnothing_4 = L_{O_4D} \cos \varnothing_3 + L_{DF} \cos \varnothing_4 \tag{15}$$

$$X_{CW2} = X_{W4} + L_{DF} \sin \varnothing_4 = L_{O_4D} \sin \varnothing_3 + L_{DF} \sin \varnothing_4 \tag{16}$$

Speed of $W_4$ and $CW_2$

By calculating the first-order derivative of Equations (13)–(16), the centroid speed of wheel 4 and caster wheel 2 can be expressed as follows:

$$\dot{Z}_{W4} = -\dot{\varnothing}_3 L_{O_4D} \sin \varnothing_3 \tag{17}$$

$$\dot{X}_{W4} = \dot{\varnothing}_3 L_{O_4D} \cos \varnothing_3 \tag{18}$$

$$\dot{Z}_{CW2} = \dot{Z}_{W4} - \dot{\varnothing}_4 L_{DF} \sin \varnothing_4 = -\dot{\varnothing}_3 L_{O_4D} \sin \varnothing_3 - \dot{\varnothing}_4 L_{DF} \sin \varnothing_4 \tag{19}$$

$$\dot{X}_{CW2} = \dot{X}_{W4} + \dot{\varnothing}_4 L_{DF} \cos \varnothing_4 = \dot{\varnothing}_3 L_{O_4D} \cos \varnothing_3 + \dot{\varnothing}_4 L_{DF} \cos \varnothing_4 \tag{20}$$

Acceleration of $W_4$ and $CW_2$

By calculating the second-order derivative of Equations (17)–(20), the centroid acceleration of wheel 4 and caster wheel 2 can be expressed as follows:

$$\ddot{Z}_{W4} = -\dot{\varnothing}_3{}^2 L_{O_4D} \cos \varnothing_3 \tag{21}$$

$$\ddot{X}_{W4} = -\dot{\varnothing}_3{}^2 L_{O_4D} \sin \varnothing_3 \tag{22}$$

$$\ddot{Z}_{CW2} = \ddot{Z}_{W4} - L_{DF} \left( \dot{\varnothing}_4{}^2 \cos \varnothing_4 \right) = -\dot{\varnothing}_3{}^2 L_{O_4D} \cos \varnothing_3 - L_{DF} \left( \dot{\varnothing}_4{}^2 \cos \varnothing_4 \right) \tag{23}$$

$$\ddot{X}_{CW2} = \ddot{X}_{W4} - L_{DF} \left( \dot{\varnothing}_4{}^2 \sin \varnothing_4 \right) = -\dot{\varnothing}_3{}^2 L_{O_4D} \sin \varnothing_3 - L_{DF} \left( \dot{\varnothing}_4{}^2 \sin \varnothing_4 \right) \tag{24}$$

Equations (21)–(24) constitute the kinematic equation for the robot driving wheel and caster wheel in climbing the wind turbine tower capability. The analysis of the $W_1$ and $CW_1$ and $W_4$ and $CW_2$ are relatively the same to $W_2$ and $CW_1$ and $W_3$ and $CW_2$.

### 2.2.3. Analysis of the Robot Body

Using the same method, the coordinate, speed, and acceleration of the center of mass ($G'$) of the robot can be obtained using Equations (25)–(30):

Position

$$Z_G = L_{O_1G'} \sin \varnothing_5 \tag{25}$$

$$X_G = L_{O_1G'} \cos \varnothing_5 \tag{26}$$

Speed

$$\dot{Z}_G = \dot{\varnothing}_5 L_{O_1G'} \cos \varnothing_5 \tag{27}$$

$$\dot{X}_G = -\dot{\varnothing}_5 L_{O_1G'} \sin \varnothing_5 \tag{28}$$

Acceleration

$$\ddot{Z}_G = -\dot{\varnothing}_5{}^2 L_{O_1G'} \sin \varnothing_5 \tag{29}$$

$$\ddot{X}_G = -\dot{\varnothing}_5{}^2 L_{O_1G'} \cos \varnothing_5 \tag{30}$$

By using these equations, the kinematic parameters of the wind-turbine-tower-climbing robot can be precisely determined.

### 2.2.4. Dynamic Analysis of Each Robot Driving Wheel

The dynamic static force analysis of the robot's wheel for a typical state of motion in the wind turbine tower surface is analyzed. The analysis process of wheel 1 is similar to that of wheel 3, as is the case with wheel 2 and wheel 4, respectively. Through this process, the forces of the wheel that constitute in all axes are being shown in in the following:

### Forces of Wheel 1

From the wheel 1 perspective, as shown in Figure 4a top view, the forces are the $F_{n1}$ normal force exerted of $W_1$; $F_{f1}$ is the frictional force of $W_1$; $F_t$ is the force from the Bowden cable to attain the ideal traction force; $\alpha_1$ is an angle of the normal force with respect to $X$-axis; $\alpha_2$ is an angle of the frictional force with respect to $X$-axis; Figure 4b shows a side view, where $\beta$ is the tower inclination angle; $\tau$ is the torque required from the DC motor; mg is the center mass of gravity and $r_1$ is the radius of the robot wheel as it rolls to the tower surface. The mechanism can be regarded as being in equilibrium state; thus, the equation for each axis can be obtained from Equations (31)–(36).

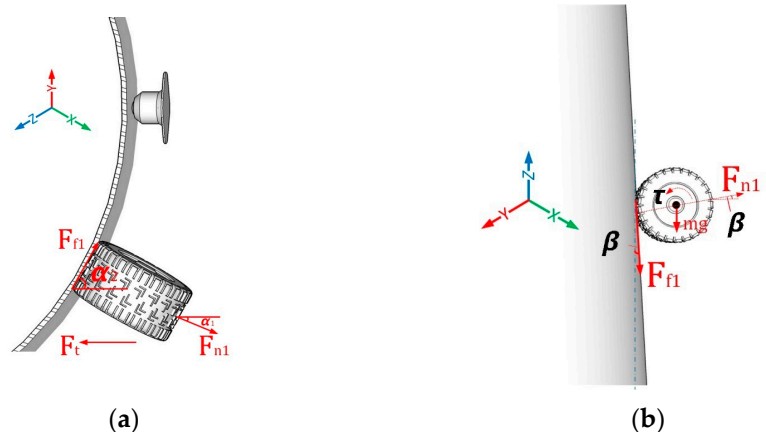

|                    |                    |
| :----------------: | :----------------: |
| **(a)**            | **(b)**            |

**Figure 4.** Wheel 1: (**a**) top view and (**b**) side view.

At $X$-axis:

$$\sum F_x = m\ddot{x} \tag{31}$$

$$F_{n1} \cos \alpha_1 + F_{f1} \cos \alpha_2 = 2F_t + m\ddot{x} \tag{32}$$

At $Y$-axis:

$$\sum F_y = 0 \tag{33}$$

$$F_{n1} \sin \alpha_1 = F_{f1} \sin \alpha_2 \tag{34}$$

At $Z$-axis:

$$\sum F_z = m\ddot{z} \tag{35}$$

$$F_{n1} \sin \beta - F_{f11} \cos \beta - m_1 g + F_D = m\ddot{z} \tag{36}$$

where $F_D$ is the force from the axel, $F_{f11} = \mu F_{n1}$ and $m_1 = M_{R1}/2$; $M_{R1}$ is the total mass of the body frame with winding.

↺ moment at W1 :

The moment at wheel 1 rotates counterclockwise to the tower surface; its contact point gives Equation (37).

$$\left(F_{f11} \cos \beta\right) r_1 + \tau = \frac{J\ddot{x}}{r_1} \tag{37}$$

Forces of Wheel 2

From the wheel 2 perspective, as shown in Figure 5a top view, the forces are the $F_{n2}$ normal force exerted of $W_2$; $F_{f2}$ is the frictional force of $W_2$; $F_t$ is the force from the Bowden cable to attain the ideal traction force; $\alpha_1$ is an angle of the normal force with respect to *X*-axis; $\alpha_2$ is an angle of the frictional force with respect to *X*-axis; Figure 5b shows side view, where $\beta$ is the tower inclination angle; $\tau$ is the torque required from the DC motor; mg is the center mass of gravity and $r_2$ is the radius of the robot wheel as it rolls to the tower surface. The mechanism can be regarded as in equilibrium state; thus, the equation for each axis can be obtained from Equations (38)–(43).

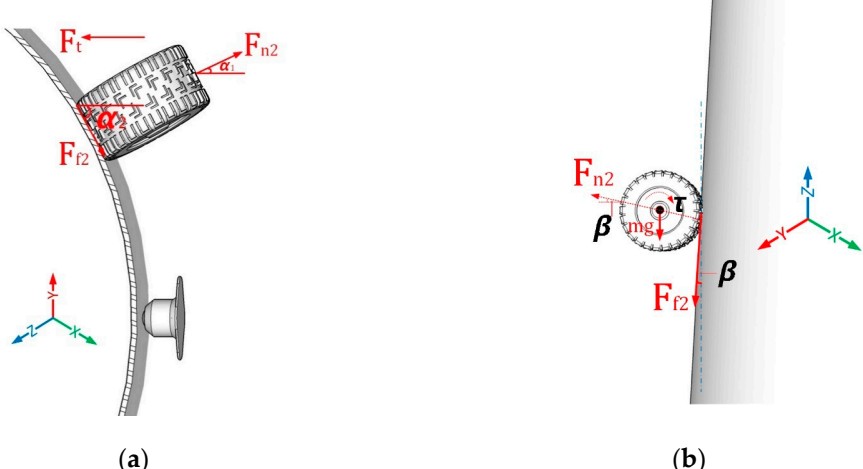

**Figure 5.** Wheel 2: (**a**) top view and (**b**) side view.

At *X*-axis:

$$\sum F_x = m\ddot{x} \tag{38}$$

$$F_{n2}\cos\alpha_1 + F_{f2}\cos\alpha_2 = 2F_t + m\ddot{x} \tag{39}$$

At *Y*-axis:

$$\sum F_y = 0 \tag{40}$$

$$F_{n2}\sin\alpha_1 = F_{f2}\sin\alpha_2 \tag{41}$$

At *Z*-axis:

$$\sum F_z = m\ddot{z} \tag{42}$$

$$F_{n2}\sin\beta - F_{f21}\cos\beta - m_2 g + F_D = m\ddot{z} \tag{43}$$

where $F_D$ is the force from the axel, $F_{f21} = \mu F_{n2}$ and $m_2 = M_{R1}/2$; $M_{R1}$ is the total mass of the body frame with winding.

  ↻   moment at W2 :

The moment at wheel 2 rotates clockwise to the tower surface; its contact point gives Equation (44).

$$\left(F_{f21}\cos\beta\right)r_1 + \tau = \frac{J\ddot{x}}{r_2} \tag{44}$$

Forces of Wheel 3

From the wheel 3 perspective, as shown in Figure 6a top view, the forces are the $F_{n3}$ normal force exerted of $W_3$; $F_{f3}$ is the frictional force of $W_3$; $F_t$ is the force from the Bowden cable to attain the ideal traction force; $\alpha_1$ is an angle of the normal force with respect to *X*-axis; $\alpha_2$ is an angle of the frictional force with respect to *X*-axis; Figure 6b shows side view, where $\beta$ is the tower inclination angle; $\tau$ is the torque required from the DC motor;

mg is the center mass of gravity and $r_3$ is the radius of the robot wheel as it rolls to the tower surface. The mechanism can be regarded as being in an equilibrium state; thus, the equation for each axis can be obtained from Equations (45)–(50).

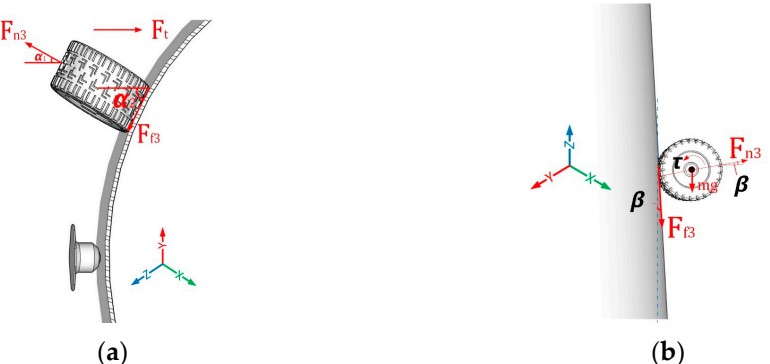

(**a**)          (**b**)

**Figure 6.** Wheel 3: (**a**) top view and (**b**) side view.

At *X*-axis:

$$\sum F_x = m\ddot{x} \tag{45}$$

$$- F_{n3} \cos \alpha_1 - F_{f3} \cos \alpha_2 = -2F_t + m\ddot{x} \tag{46}$$

At *Y*-axis:

$$\sum F_y = 0 \tag{47}$$

$$F_{n3} \sin \alpha_1 = F_{f3} \sin \alpha_2 \tag{48}$$

At *Z*-axis:

$$\sum F_z = m\ddot{z} \tag{49}$$

$$F_{n3} \sin \beta - F_{f31} \cos \beta - m_3 g + F_D = m\ddot{z} \tag{50}$$

where $F_D$ is the force from the axel, $F_{f31} = \mu F_{n3}$ and $m_3 = M_{R2}/2$; $M_{R2}$ is the total mass of the body frame without winding.

↬    moment at W3 :

The moment at wheel 3 rotates counterclockwise to the tower surface; its contact point gives Equation (51).

$$\left( F_{f31} \cos \beta \right) r_3 + \tau = \frac{J\ddot{x}}{r_3} \tag{51}$$

Forces of Wheel 4

From the wheel 4 perspective, as shown in Figure 7a top view, the forces are the $F_{n4}$ normal force exerted of $W_4$; $F_{f4}$ is the frictional force of $W_4$; $F_t$ is the force from the Bowden cable to attain the ideal traction force; $\alpha_1$ is an angle of the normal force with respect to *X*-axis; $\alpha_2$ is an angle of the frictional force with respect to *X*-axis; Figure 7b shows side view, where $\beta$ is the tower inclination angle; $\tau$ is the torque required from the DC motor; mg is the center mass of gravity and $r_2$ is the radius of the robot wheel as it rolls to the tower surface. The mechanism can be regarded as being in an equilibrium state; thus, the equation for each axis can be obtained from Equations (52)–(57).

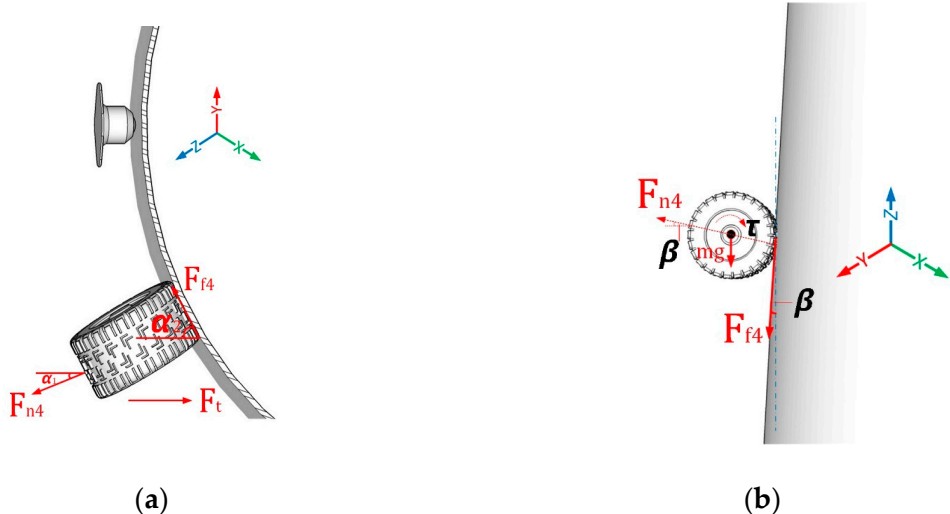

**Figure 7.** Wheel 4: (**a**) top view and (**b**) side view.

At *X*-axis:

$$\sum F_x = m\ddot{x} \tag{52}$$

$$-F_{n4}\cos\alpha_1 - F_{f4}\cos\alpha_2 = -2F_t + m\ddot{x} \tag{53}$$

At *Y*-axis:

$$\sum F_y = 0 \tag{54}$$

$$F_{n4}\sin\alpha_1 = F_{f4}\sin\alpha_2 \tag{55}$$

At *Z*-axis:

$$\sum F_z = m\ddot{z} \tag{56}$$

$$F_{n4}\sin\beta - F_{f41}\cos\beta - m_4 g + F_D = m\ddot{z} \tag{57}$$

where $F_D$ is the force from the axel, $F_{f41} = \mu F_{n4}$ and $m_4 = M_{R2}/2$; $M_{R2}$ is the total mass of the body frame without winding.

∽ moment at W4 :

The moment at wheel 4 rotates clockwise to the tower surface; its contact point gives Equation (58).

$$\left(F_{f41}\cos\beta\right)r_4 + \tau = \frac{J\ddot{x}}{r_4} \tag{58}$$

Forces on Caster Wheel 1 with Winding

From the caster wheel 1 perspective, as shown in Figure 8a top view, the forces are the $F_{ncw1}$ normal forces exerted of $CW_1$; $F_{BF1}$ is the force of the body frame 1 weight exerted to the tower surface; $F_t$ is the force from the Bowden cable to attain the ideal tension force; Figure 8b shows side view, where $\beta$ is the tower inclination angle; mg is the center mass of gravity and $\gamma$ is an angle caused by the $BF_1$ weight. The mechanism can be regarded as being in an equilibrium state; thus, the equation for each axis can be obtained from Equations (59)–(63).

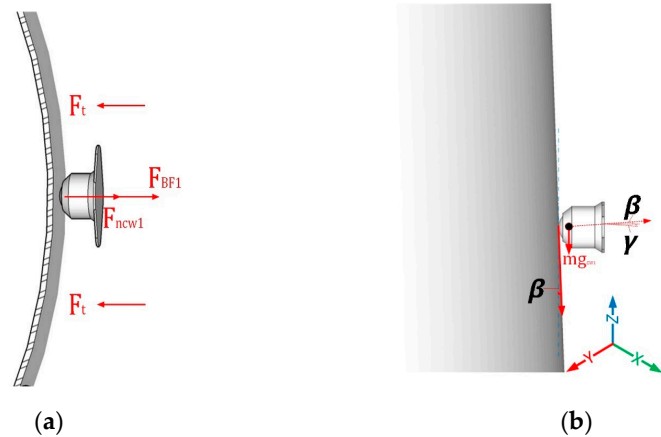

**Figure 8.** Caster Wheel 1: (**a**) top view and (**b**) side view.

At *X*-axis:

$$\sum F_x = m\ddot{x} \tag{59}$$

$$2F_t - F_{ncw1} - F_{BF1} \cos\gamma = m\ddot{x} \tag{60}$$

At *Y*-axis:

$$\sum F_y = 0 \tag{61}$$

At *Z*-axis:

$$\sum F_z = m\ddot{z} \tag{62}$$

$$F_{ncw1}\sin\beta - F_{fcw1}\cos\beta - m_{cw1}g - F_{BF1}\cos\gamma = m\ddot{z} \tag{63}$$

where $F_{fcw1} = \mu F_{ncw1}$
$F_{BF1}$ = Force of Body Frame 1 with winding
$F_{BF2}$ = Force of Body Frame 2 without winding

Forces on Caster Wheel 2 without Winding

From the caster wheel 2 perspective, as shown in Figure 9a top view, the forces are the $F_{ncw2}$ normal forces applied exerted of $CW_2$; $F_{BF2}$ is the force of the body frame 1 weight exerted to the tower surface; $F_t$ is the force from the Bowden cable to attain the ideal tension force; Figure 9b shows side view, where $\beta$ is the tower inclination angle; mg is the center mass of gravity and $\gamma$ is an angle caused by the $BF_2$ weight. The mechanism can be regarded as being in an equilibrium state; thus, the equation for each axis can be obtained from Equations (64)–(68).

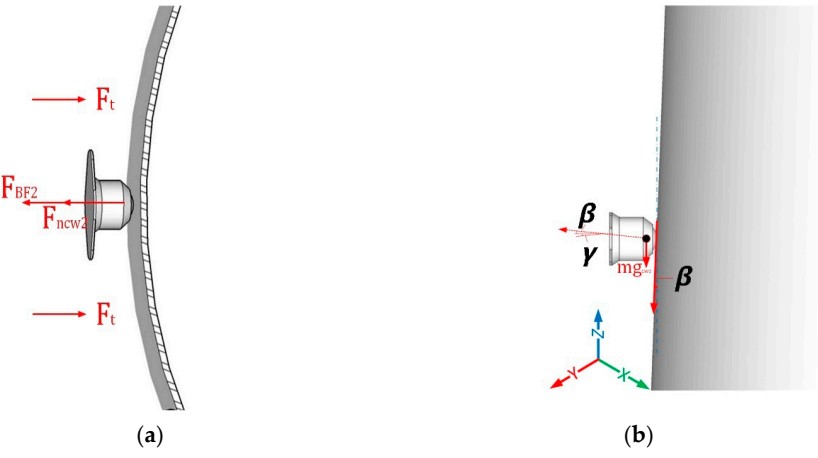

**Figure 9.** Caster Wheel 2: (**a**) top view and (**b**) side view.

At *X*-axis:

$$\sum F_x = m\ddot{x} \tag{64}$$

$$2F_t - F_{ncw2} - F_{BF2}\cos\gamma = m\ddot{x} \tag{65}$$

At *Y*-axis:

$$\sum F_y = 0 \tag{66}$$

At *Z*-axis:

$$\sum F_z = m\ddot{z} \tag{67}$$

$$F_{ncw2}\sin\beta - F_{fcw2}\cos\beta - m_{cw2}g - F_{BF2}\sin\gamma = m\ddot{z} \tag{68}$$

where $F_{fcw2} = \mu F_{ncw2}$

### *2.3. Control System and Experimental Set-Up*

In this context, the details on the control system used in the prototype, control flowchart of the climbing robot and experimental set-up of the wind-turbine-tower-climbing robot were discussed.

#### 2.3.1. Control System

The control system of the wind-turbine-tower-climbing robot's hardware is shown in Figure 10. The main microcontroller is the Arduino MEGA 2560, which incorporates all the up–down movement controls and carries out the tasks of other electronic modules to harmonize the movement process with the various speeds. This microcontroller controls the four driving wheel motors of the robot, one step motor for the winding mechanism, distance sensor, motor encoder, current sensor, and other support electronic components. It is also considered that, due to its small size, inexpensive cost, reliability, and availability in the market, this microcontroller is good enough for the development of the wind-turbine-tower-climbing robot as it requires a straightforward control procedure and minimum programing code. This study utilized an open-source programming tool called Integrated Development Environment (IDE) to program Arduino boards [26], which supports C-programming code of the wind-turbine-tower-climbing robot, as shown in Figure 11, and then moved to the Arduino board using a USB cable. Its programmable input/output (I/O) pins are configured using C++ programming software, which includes pulse width modulation (PWM) pins and analogue pins. PWM pins can be used to control four DC motors and one step motor, while analogue pins have analogue-to-digital converter (ADC), which can be used to read signals from the distance sensor. The ultrasonic distance sensor—HCSR04 automatically provides feedback signal measurement ranging 200 mm to reach mock-up tower height of 1200mm to the Arduino microcontroller to attain the precise adhesion in the wind turbine tower mock-up without slipping to the ground. After the distance sensor was able to detect signal the winding mechanism was the next process. With the help of one step motor-42HS03 with twin pulley attached to its shaft, the Bowden cable, in terms of winding and unwinding, achieves the precise adhesion between the two-body frame in the tower. The step motor slowly winds or rolls-in its cable as it climbs to the tower in an upward movement or unwinds or rolls-out in a downward movement. The support electronics module is needed to run the other devices, such as encoders, while some were required for acquiring data during the entire movement process. A regulated 5 V DC voltage source was needed to power up the Arduino microcontroller and other boards. Once everything was in the right position, the four driving wheels GP-BLDC3650 will push the entire robot in a straight upward and downward movement along the wind turbine tower mock-up. A 12 V DC power supply to the four motors are needed to drive the whole robot.

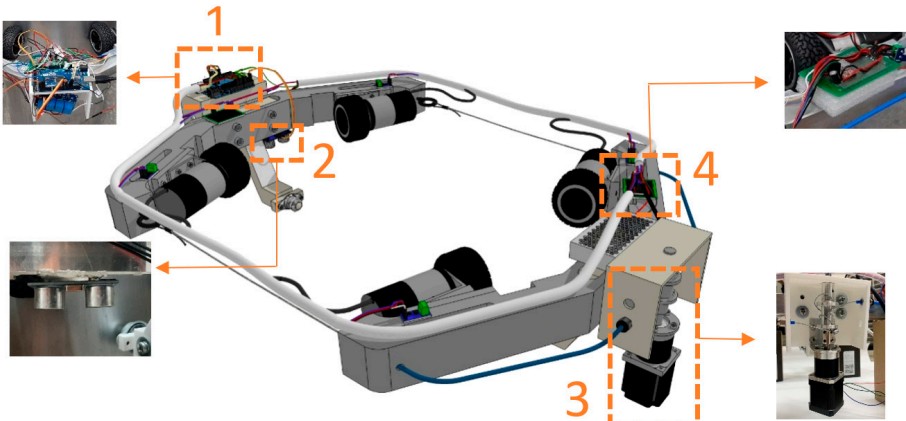

**Figure 10.** Control system of the robot: (**1**) Arduino MEGA 2560; (**2**) Ultrasonic Distance Sensor—HC-SR04; (**3**) Motor Encoder; (**4**) Step Motor with twin pulley.

```
Pseudocode for Climbing Robot
    Procedure Input_Speed(Mspeed)
        Switch Mspeed
        Case Low:        Mspeed = A mm/s
        Case Medium:  Mspeed = B mm/s
        Case High:      Mspeed = C mm/s
        End switch
        goto Distance_measure
    end procedure
    Procedure Distance_Measure(D)
        D = sensor_reading
        if D < 1200mm
                goto Ascending
        else if D > 80mm
                goto Descending
        else STOP_Program
        endif
    end procedure
    Procedure Ascending
        Mwind(rotation) = CW
        M1wheel(speed) = Mspeed
        M2wheel(speed) = Mspeed
        M3wheel(speed) = Mspeed
        M4wheel(speed) = Mspeed
        goto Distance_measure
        goto interrupt_check
    end procedure
    Procedure Descending
        Mwind(rotation) = CCW
        M1wheel(speed) = -Mspeed
        M2wheel(speed) = -Mspeed
        M3wheel(speed) = -Mspeed
        M4wheel(speed) = -Mspeed
        goto Distance_measure
        goto interrupt_check
    end procedure
```

**Figure 11.** The C++ language of wind-turbine-tower-climbing robot.

### 2.3.2. Control Flowchart of the Wind-Turbine-Tower-Climbing Robot

The flowchart in Figure 12 illustrates the climbing algorithm of the wind-turbine-tower-climbing robot. As the system starts, implying that all the hardware and software are turned on, the system will have its proper initialization to set up all the resources in the system to function effectively, specifically the driving and winding mechanism. The user or the operator had the ability to choose between the various speed options: Low speed, which takes 20 s, Medium speed, which takes 16 s, and Fast speed, which takes 8 s for the robot to finish the climbing process. The input selection of the various speeds is necessary for the ultrasonic distance sensor to activate and measure the distance from the

ground; either the distance is less than or equal to 80 mm or the distance is greater than 80 mm. If the distance was less than or equal to 80 mm, the step motor would execute the winding or tightening of the Bowden cable so that the robot would adhere around the tower surface. Once the correct distance was attained, the four rubber wheels as the locomotion mechanism would automatically roll up to the tower surface, and if there was no interruption, it would continuously do the same process until it reaches the distance of 1200 mm. It will then end at "B", which means the robot completed the straight up movement to the tower surface and was ready to roll down. The robot would continue to move down the tower until the measured distance reached less than 80mm. If the initially measured distance was greater than 80 mm, the user or the operator had to input the direction of the climbing robot, that is, whether it should go up or down. In the situation that the climbing robot would roll down to the tower, the step motor would execute the unwinding or loosening of the Bowden cable, and the four rubber wheels would move outwardly so that the robot would automatically roll down the tower surface. If there was no interruption, it would continuously do the same process until it reaches a distance of less than 80 mm, then end the process. However, if the distance is not attained, it will continuously unwind or loosen the Bowden cable. The other way around will lead to "A", which means the robot will have to do the winding or tightening process. The robot also could stop once the manual stop was initiated or during any wind speed disturbances.

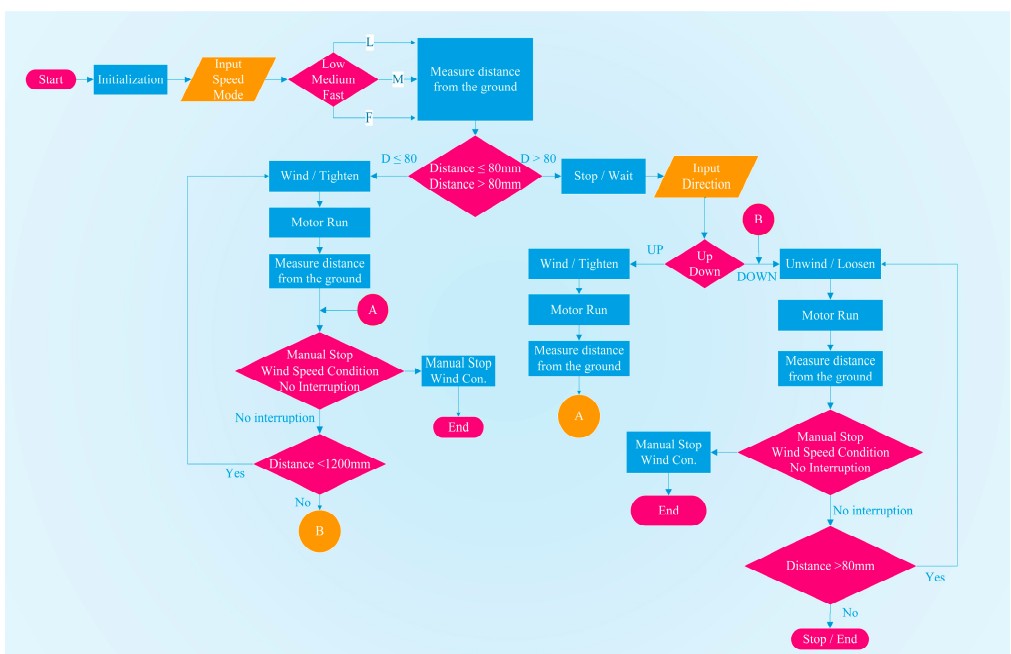

**Figure 12.** Control flowchart.

### 2.3.3. Experimental Set-Up

An experimental setup of wind-turbine-tower-climbing robot has been established in Southern Taiwan University of Science and Technology shown in Figures 13a, 14a, 15a and 16a through the CAD design alongside the final prototype Figures 11b, 12b, 13b and 14b during the actual indoor experiments with the wind turbine tower mock-up, as the realization of the lower dimension from 2 MW wind turbine tower. Figure 13b depicts the actual physical view of the climbing robot attached to the tower, Figure 14b shows the top view of the four rubber wheels as the driving mechanism and the two-caster wheels, and the two body frames are made of aluminum alloy 6061 material, which is low in weight and economical to use. The body frame 1 holds the winding mechanism, power supply of the robot and other electronics modules, as shown in Figure 15b, and the body frame 2 holds

the Arduino microcontroller, distance sensors, relays, encoders, and support electronics devices, as shown in Figure 16b.

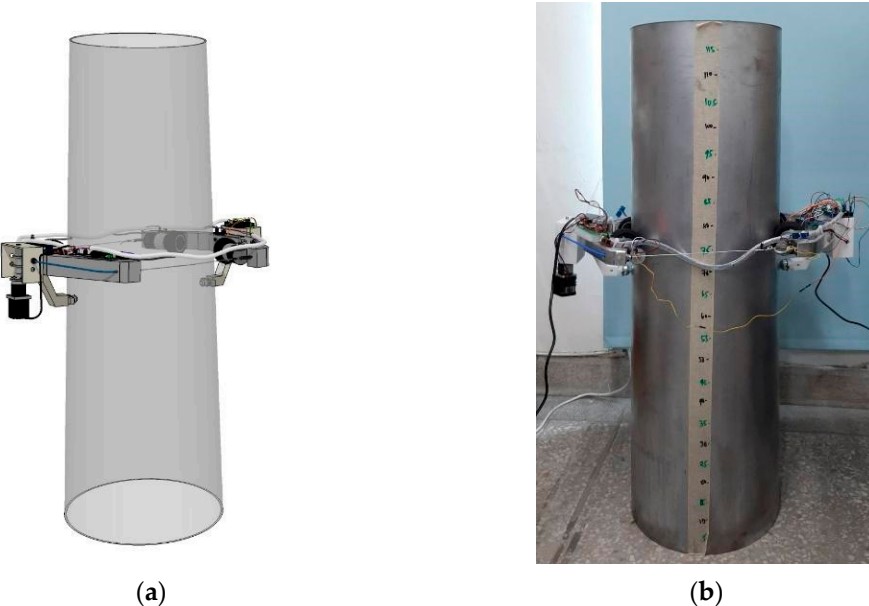

(**a**)　　　　　　　　　　　　　　　　　(**b**)

**Figure 13.** Wind-turbine-tower-climbing robot: (**a**) CAD design and (**b**) actual physical view.

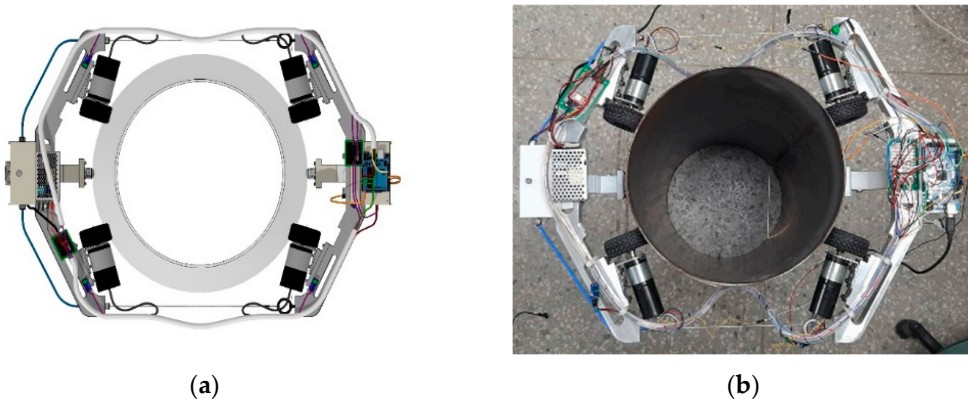

(**a**)　　　　　　　　　　　　　　　　　(**b**)

**Figure 14.** Top view: (**a**) CAD design and (**b**) actual physical view.

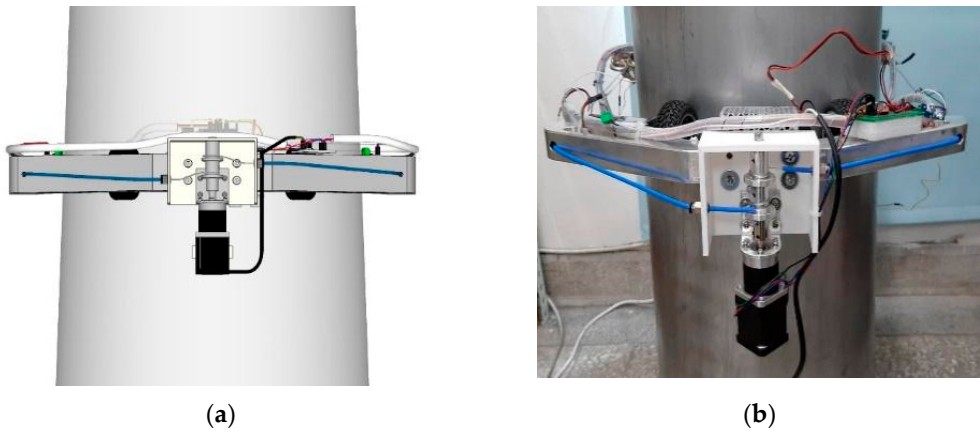

(**a**)　　　　　　　　　　　　　　　　　(**b**)

**Figure 15.** Body Frame 1: (**a**) CAD design and (**b**) actual physical view.

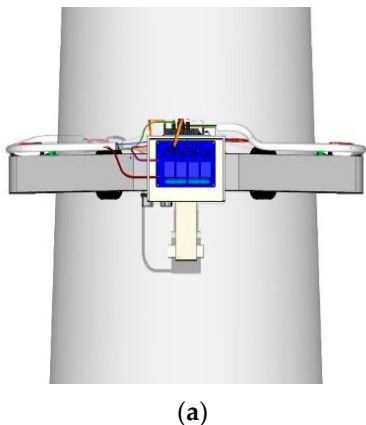
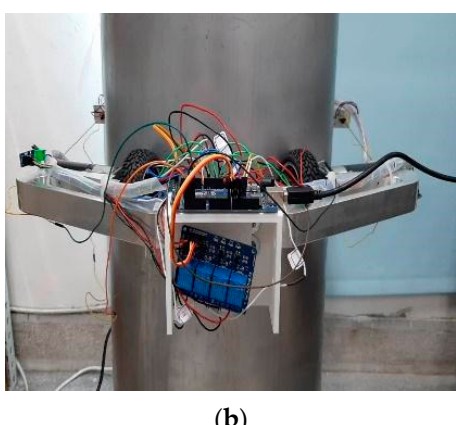

(**a**)                                                            (**b**)

**Figure 16.** Body Frame 2: (**a**) graphic design and (**b**) actual physical view.

The two main mechanisms for the wind-turbine-tower-climbing robot were the locomotion and the adhesion mechanism. The locomotion comprises the four brushless DC planetary gear motorized rubber wheels, which have a 7–30-watt rating power and 80–100 round per minute rating speed, and two caster wheels used to support the body frame of the robot. Next, the adhesion mechanism uses a winding cable that is triggered from the distance sensor to attain the precise tension force to ensure enough adhesion to the tower circumference for climbing. From the distance sensor, an automatic setting in the microcontroller program was established to achieve the step motor speed adjustments in the winding and unwinding process of the cable and to have a synchronization with the four driving wheels ready for the wind turbine tower movement. Because there was only one step motor installed in the body frame of the robot, it was easy to control the cable adjustment to alter the distance between the tower center. When the step motor with twin pulley winds the Bowden cable inwardly, the robot body frame would tighten to the tower surface, the rubber wheels would contact the tower area and the corresponding normal force would increase. As a result, the normal forces provide the frictional force by multiplying the friction coefficient together with the angle of inclination of the tower, which overcomes the robot gravity force to make the robot climb straight up. When the step motor with twin pulley unwinds, the Bowden cable moves outwardly, causing the robot body frame to loosen to the tower surface, and the robot automatically goes straight down. The step motor gives good low speed performance in winding and unwinding the Bowden cable, and the winding rpm speed is around 19–25 rpm. Once the precise measurement in the adhesion mechanism was achieved from the distance sensor, the robot starts its movement on the tower. The robot needs 3 s to start the upward movement and another 3 s to start the downward movement of the tower height of 1200 mm.

To verify the payload capacity of the prototype, as shown in Figure 17a,b, approximately 3.540 kg was implemented to test the load capacity of the robot. The results indicate that the climbing robot was able to climb straight up and down in the wind turbine tower mock-up regardless of its various speeds of climbing, whether low, medium, or fast, as well as the different tower diameters. The various payload capacities of the robot are 0.823 kg per body frame, with a total of 1.646 kg in the trial 1 test load, adding 0.947 kg per body frame, with a total of 3.540 kg in the trial 2 test load; regardless of the climbing speed of the robot, its climbing ability has no changes. The robot can withstand the maximum payload weight of 5.44 kg that requires high torque and velocity decrease. During the indoor experiment, the robot could handle climbing its maximum payload weight but, due to the light material that supported the body frame attached to the caster wheel, it tends to be the first to break after several instances of climbing. In this condition, reaching the 50% payload capacity from the robot's total weight would be the optimum load that the robot can hold regardless of various climbing speeds. On the other hand, the four driving wheels

did not slip from the total weight of the robot, which is 6.56 kg, signifying that the frictional force of the driving wheel satisfied the climbing condition from the various speeds.

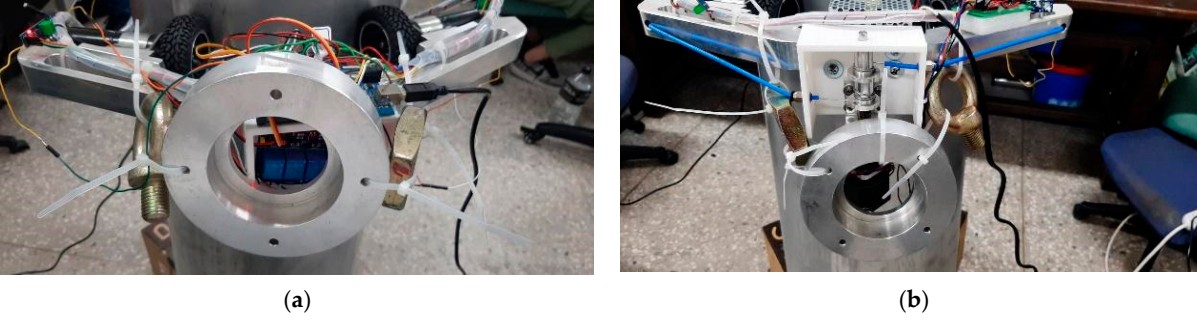

**Figure 17.** Payload capacity: (**a**) Body Frame 1 and (**b**) Body Frame 2.

## 3. Results and Discussion

A wind-turbine-tower-climbing robot was designed and implemented through an actual indoor experiment involving the tower mock-up to assess how effectively the performance of the driving and winding mechanism works in the various speeds and payload capacity. From the indoor testing, there is a 200 mm baseline due to robot height. The total distance for the robot to climb is roughly 1000 mm from the maximum tower height of 1200 mm mock-up from the 2 MW wind turbine tower.

### 3.1. Experimental Results of Distance and Time

When the robot was climbing in the upward movement, the actual data on distance versus time and speed versus time were taken. The experiment results in various speeds: Low speed, which takes 20 s, Medium speed which takes 16 s, and Fast speed which takes 8 s to climb up the tower. The 3540 kg payload that the robot can hold without any problem at the various speeds shown in Figures 18–20 is included in the climbing process.

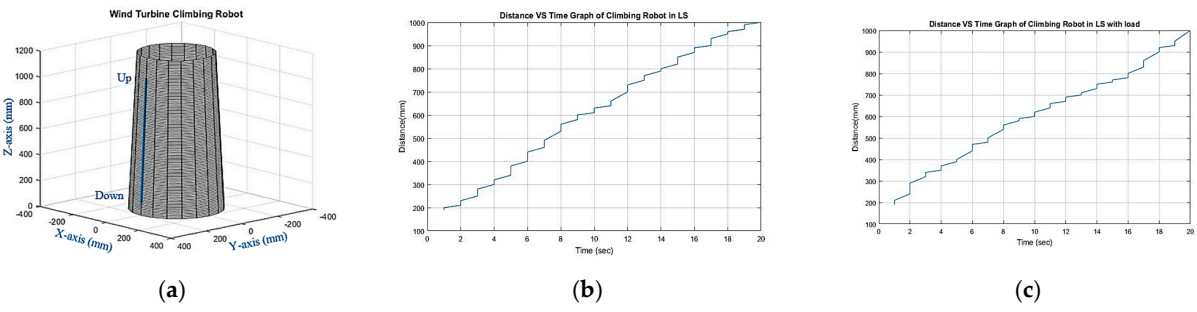

**Figure 18.** Distance and Time: (**a**) Low Speed (LS) at 20 s, (**b**) distance vs. time in LS and (**c**) distance vs. time in LS with load.

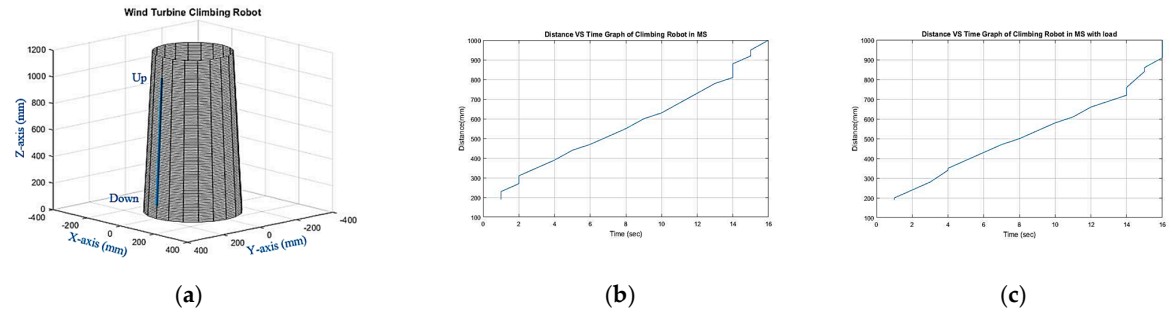

**Figure 19.** Distance and Time: (**a**) Medium Speed (MS) at 16 s, (**b**) distance vs. time in MS and (**c**) distance vs. time in MS with load.

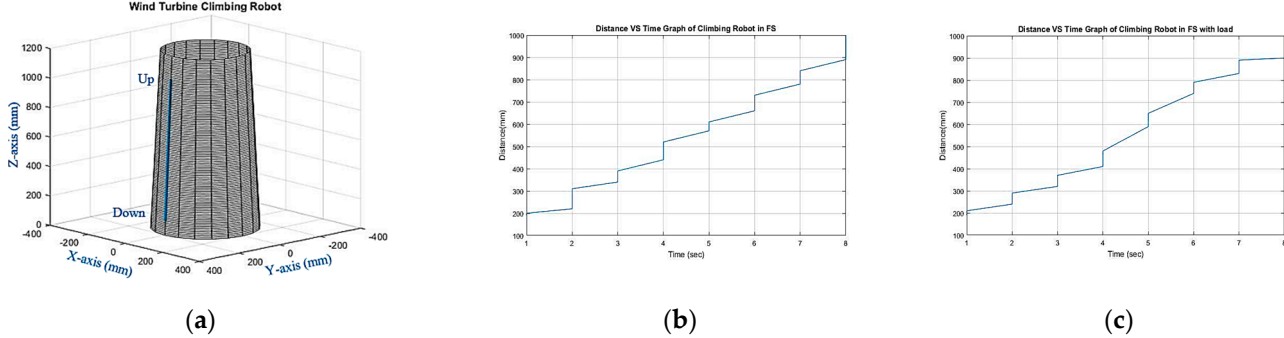

(a)                                    (b)                                    (c)

**Figure 20.** Distance and Time: (**a**) Fast Speed (FS) at 8 s, (**b**) distance vs. time in FS and (**c**) distance vs. time in FS with load.

Based on Figures 18a, 19a and 20a, its distance versus time results during the indoor experiment are as follows: the different durations of time are 20, 16, and 8 s to climb straight up to the 1200 mm wind turbine tower height and approximately 1000 mm distance traveled since the robot has a height of 200 mm. The distance sensor is connected to the robot's body frame and can detect the robot's distance in the tower. The distance was displayed on the user monitor in terms of centimeters, which is then converted to millimeters. The first climbing speed was Low at 20 s; there was no large disparity between the test results of "without load" and "with load at 3540 kg," as shown in Figure 18b,c, the two graphs of distance vs. time in LS. This means that the robot can handle the movement going straight up the tower surface, and in a second there are two quick variations on its constant speed. The second climbing speed was Medium at 16 s; there is no large disparity between the test results of "without load" and "with load at 3540 kg", as shown in Figure 19b,c, the two graphs of distance vs. time in MS. Still, the robot can handle the movement going straight up the tower surface at a smooth and steady speed for almost the entire duration. The third climbing speed was Fast at 8 s; there is no large disparity between the test results of "without load" and "with load at 3540 kg", as shown in Figure 20b,c, the two graphs of distance vs. time in FS. The robot can handle the movement going straight up the tower surface, and every second there is change on its constant speed. From the test results of the robot climbing in the three various speeds the robot can suffice to climb straight up movement without breaking or slipping onto the ground. Its downward movement was also the same time and speed to roll out to the wind turbine tower mock-up. The actual tests of the climbing robot without and with payload are shown in Figures 21 and 22.

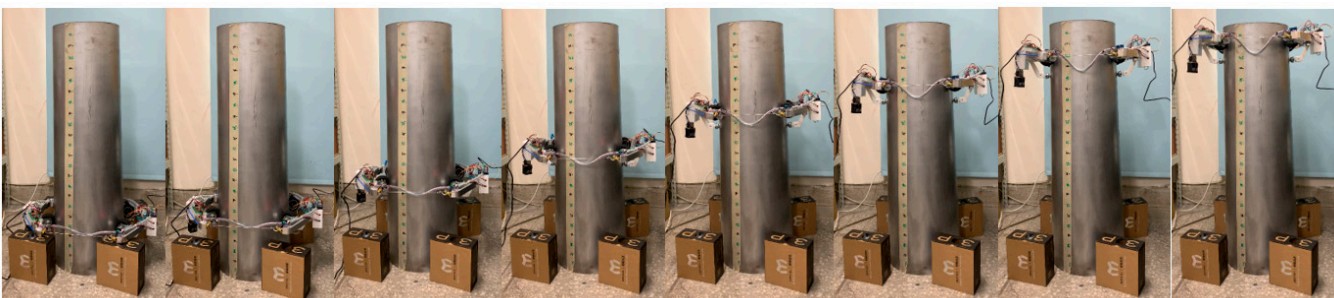

**Figure 21.** Climbing robot without payload.

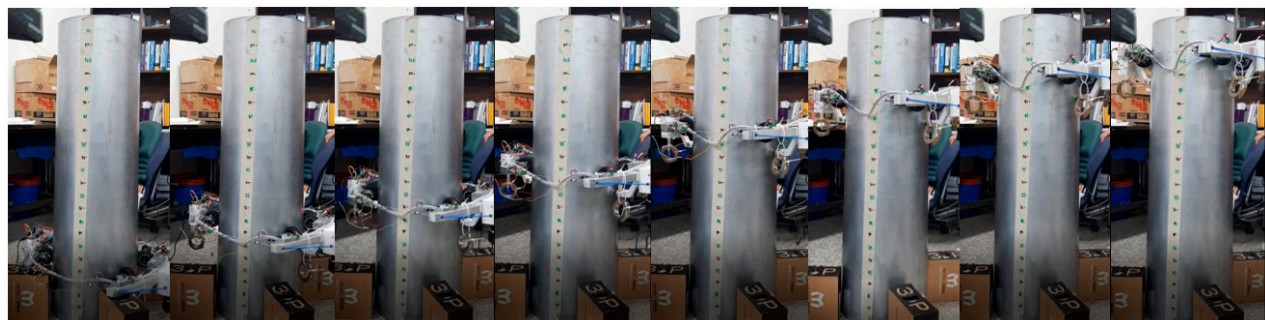

**Figure 22.** Climbing robot with payload.

### 3.2. Experimental Results of (Speed) RPM and Time

Based on Figures 23a, 24a and 25a, speed versus time results during the indoor experiment included the speed of four driving wheels in terms of rpm (revolution per minute) versus the different durations of time, which are 20, 16, and 8 s. The first climbing speed was low at 20 s, as shown in Figure 23b,c from the two graphs' test results for "without load" and "with load of 3.540 kg". The rpm of each wheel moved in a subsequent fluctuation that is around 45–57 and 59–74 rpm, respectively. The second climbing speed was Medium at 16 s; as shown in Figure 24b,c, the rpm of each wheel moved in a subsequent fluctuation, around 44–52 and 44–61 rpm, respectively. The third climbing speed was Fast at 8 s; as shown in Figure 25b,c, the rpm of each wheel moved in a subsequent fluctuation, around 44–54 and 52–77 rpm, respectively. Based on the graph, it is obvious that "with load" the rpm in each motor wheel of the robot was powerful enough to overcome the load torque and force that prevents movement. The detailed specifications are listed in Table 2.

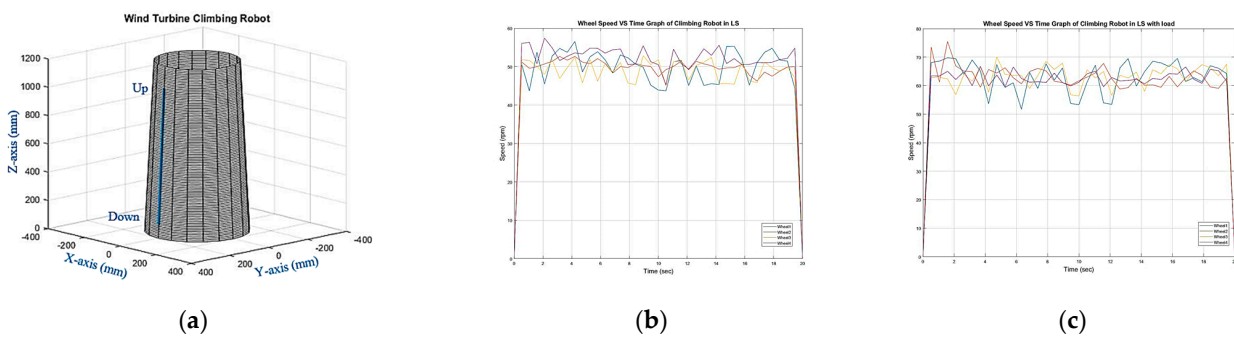

|   |   |   |
|---|---|---|
| (a) | (b) | (c) |

**Figure 23.** RPM and time: (**a**) Low Speed (LS) at 20 s, (**b**) rpm vs. time in LS and (**c**) rpm vs. time in LS with load.

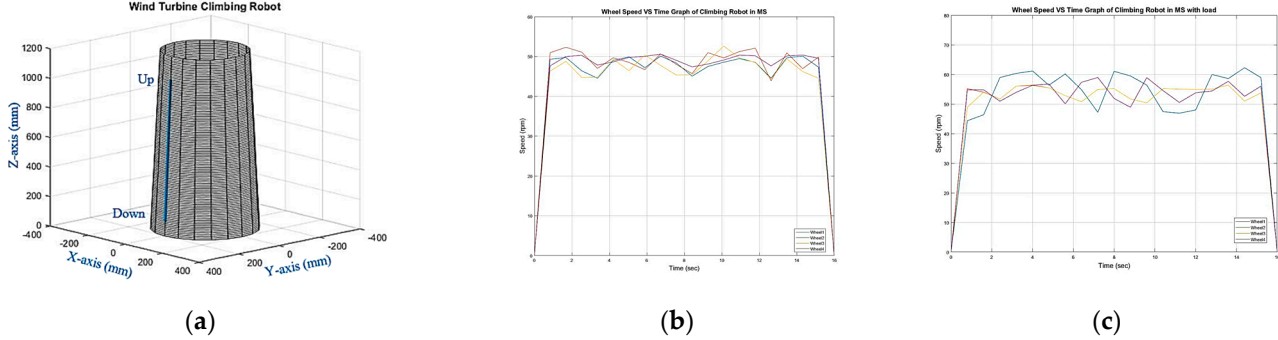

|   |   |   |
|---|---|---|
| (a) | (b) | (c) |

**Figure 24.** RPM and time: (**a**) Medium Speed (MS) at 16 s, (**b**) rpm vs. Time in MS and (**c**) rpm vs. Time in MS with load.

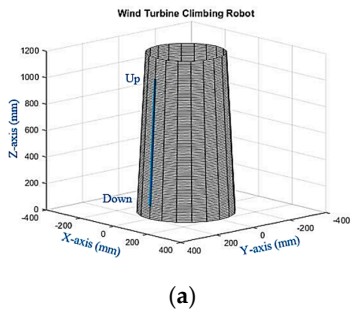
(**a**)

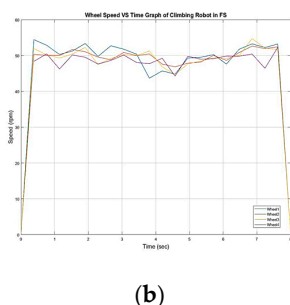
(**b**)

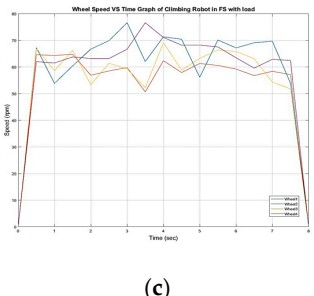
(**c**)

**Figure 25.** RPM and time: (**a**) Fast Speed (FS) at 8 s, (**b**) rpm vs. time in FS and (**c**) rpm vs. time in FS with load.

**Table 2.** Wind-turbine-tower-climbing robot specifications.

| Specifications | | Values | |
|---|---|---|---|
| Robot Weight | *Body frame 1* | 6.56 kg | 3.64 kg |
| | *Body frame 2* | | 2.92 kg |
| Max payload weight | | 3.540 kg | |
| Average Speed | *Low* | 50 mm/s | |
| | *Medium* | 62.5 mm/s | |
| | *High* | 125 mm/s | |
| Wheel rpm | *Low* | 45~74 rpm | |
| | *Medium* | 44~66 rpm | |
| | *High* | 44~77 rpm | |
| Winding rpm | | 19~25 rpm | |
| Angle of inclination | | 1.67° | |
| Tower height | | 1200 mm | |
| Tower diameter for climbing | | 330~400 mm | |

## 4. Conclusions and Future Works

In this paper, the development of a wind-turbine-tower-climbing robot has been presented, along with the kinematic and dynamic analysis of each robot driving wheel, which was successfully tested through indoor experiments using the prototype with the wind turbine tower mock-up, proving the viability of the design. Based on the friction created by four rubber wheels, the robot could climb in a straight up-and-down movement at its various speeds. The winding mechanism was the significant advantage that can withstand the cable tension force to hold on to the tower surface and the high payload capacity to install additional equipment that is approximately above 50% of the robot total weight. The various wheel speeds of the robot to move along the tower mock-up were demonstrated by the indoor experimental results "without load" and "with load of 3.540 kg". Finally, the results indicate the effectiveness of the driving and winding mechanism to climb at the various speeds with and without the payload. The prototype needed to be further refined and tested, especially for use in an outdoor facility with varying wind strength. Later work included the use of wind sensors for wind gusts, a camera to observe external defects, and a robotic arm to transport inspection tools, etc. This prototype wind-turbine-tower-climbing robot could provide a testing ground for new developments.

**Author Contributions:** Conceptualization: J.-H.L. and K.E.P.; data collection: K.E.P.; theoretical analysis: J.-H.L.; experiment and test: K.E.P.; writing—original draft: J.-H.L. and K.E.P.; writing—review and editing: J.-H.L. and K.E.P. All authors have read and agreed to the published version of the manuscript.

**Funding:** This research received no external funding.

**Institutional Review Board Statement:** Not applicable.

**Informed Consent Statement:** Not applicable.

**Data Availability Statement:** Data sharing not applicable.

**Conflicts of Interest:** The authors declare no conflict of interest.

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
