# Peer review of "A Wind-Turbine-Tower-Climbing Robot Prototype Operating at Various Speeds and Payload Capacity: Development and Validation"

_applsci, doi:10.3390/app13031381_

Round 1
Reviewer 1 Report
You have presented a suitable and efficient idea But there are some points that seem to need to be reviewed There is no information about the codes related to the control unit, and the text related to the control unit lacks technical information of the project Provided a more interesting, yet balanced discussion of the study’s results Improved the paper’s framing with management theory Please do more research on the optimization methods of Arduino to new models because it will improve the productivity and develop your project.
Author Response
Thank you for giving us the opportunity to submit a revised draft of the manuscript “applsci-2155255 - “Development and Validation of a Wind Turbine Tower Climbing Robot Subject to Payload Capacity” for publication. We appreciate the time and effort for providing us feedback on our manuscript and are grateful for the insightful comments and valuable improvements to our paper. Those changes are highlighted within the manuscript. Please see below, in blue, for a point-by-point response to your comments and suggestions. All line numbers refer to the revised manuscript file with tracked changes of color RED -1st that highlighted the reviewers' comments and suggestions.

Reviewer 2 Report
This work discusses a development of a wind turbine tower climbing robot subject to payload capacity in order to assess the service lifespan of wind turbine components. The paper represents good work. However, some points should be taken into consideration before acceptance:
· Wind turbine climbing robots facilitate the digitization of inspection and maintenance operation to prolong the service lifespan and to avoid unexpected external failures of the wind turbine parts [5] such as corrosion, cracks, paint peel-off, material degradation, lightning strike damage and other physical defects [3], [6] as listed Table 1 the different physical defects occurring in the structure of wind turbine. It is very long sentence, please split it.
· The figures presented in pages 14 and 15 should have numbers and captions.
· I think the control system and experimental set-up should be in a separate section. Then followed by results and discussion.
· The figures presented in Tables 3 and 4 are very poor in terms of resolution and readability.
· In addition, tables 3 and 4 contain only figures, which is somehow strange. Typically, results are presented in the form of figures with a number and caption.
· The authors said that “After reviewing the wind turbine climbing robot application with its factors that need to be considered such as limitations on budget, materials, and time. It is recommended to implement a modular body structure with compact design into the wind turbine tower climbing robot design and its development”. However, there is no comparison between the proposed design and the designs existing in the literature
· The references should be up-to-date. However, there is only one reference in the year of 2022.

Author Response

(The authors gave the same response as above.)

Round 2
Reviewer 1 Report
I must say that the name of the article does not represent the actual function and has a similarity in name and function that will cause misunderstanding.
The text has grammatical and writing errors that should be corrected to be better in some cases, especially to explain the function of the summary writing system
Author Response
Thank you for giving us the opportunity to submit a revised draft of the manuscript “applsci-2155255 - “Development and Validation of a Wind Turbine Tower Climbing Robot Subject to Payload Capacity” for publication. We appreciate the time and effort for providing us feedback on our manuscript and are grateful for the insightful comments and valuable improvements to our paper. Those changes are highlighted within the manuscript. Please see below, in blue, for a point-by-point response to your comments and suggestions. All line numbers refer to the revised manuscript file with tracked changes of color RED -1st revision and GREEN-2nd revision that highlighted the reviewers' comments and suggestions.
